# Rate volatility and asymmetric segregation diversify mutation burden in cells with mutator alleles

Ian T. Dowsett [1], Jessica L. Sneeden[1], Branden J. Olson [2,3], Jill McKay-Fleisch[1], Emma McAuley[1], Scott R. Kennedy [1] & Alan J. Herr [1✉]

Mutations that compromise mismatch repair (MMR) or DNA polymerase ε or δ exonuclease domains produce mutator phenotypes capable of fueling cancer evolution. Here, we investigate how combined defects in these pathways expands genetic heterogeneity in cells of the budding yeast, *Saccharomyces cerevisiae*, using a single-cell resolution approach that tallies all mutations arising from individual divisions. The distribution of replication errors present in mother cells after the initial S-phase was broader than expected for a single uniform mutation rate across all cell divisions, consistent with volatility of the mutator phenotype. The number of mismatches that then segregated to the mother and daughter cells co-varied, suggesting that each division is governed by a different underlying genome-wide mutation rate. The distribution of mutations that individual cells inherit after the second S-phase is further broadened by the sequential actions of semiconservative replication and mitotic segregation of chromosomes. Modeling suggests that this asymmetric segregation may diversify mutation burden in mutator-driven tumors.

[1] Department of Laboratory Medicine and Pathology, University of Washington, Seattle, WA 98195-7705, USA. [2] Department of Statistics, University of Washington, Seattle, WA 98195-7705, USA. [3] Computational Biology Program, Fred Hutchinson Cancer Research Center, Seattle, WA 98109, USA. ✉email: alanherr@uw.edu

All tumors contain genetically divergent cells spawned by the evolutionary processes of mutation and selection. In some tumors, genetic heterogeneity arises from a "mutator phenotype"[1] due to mismatch repair (MMR) defects[2] or heterozygous exonuclease domain mutations (EDMs) affecting the leading or lagging strand DNA polymerases (pol), Polε or Polδ[3–9]. Since MMR corrects polymerase errors, when MMR and EDM mutations occur together they produce a dramatic increase in the number of unrepaired polymerase errors. The resulting tumors rapidly evolve and possess "ultramutated" genomes. Yet a full understanding of the relative contributions of mutagenesis and selection to the rise of heterogeneity within these tumors remains elusive, since cells with more mutations tend to adapt more readily.

A key unanswered question is whether the mutation rate is constant within populations of cells with mutator alleles (mutator cells). The two most common ways of measuring mutation rates are fluctuation analysis[10] and mutation accumulation lines[11]. Both assume a uniform mutation rate and report the average of hundreds or thousands of cell divisions. However, in recent years, evidence has emerged that mutagenic processes may vary from one division to the next. Kataegis and chromothripsis, for instance, sharply increase mutation burden in a single-cell division[12–14]. Indirect evidence for highly mutagenic sub-populations of cells also comes from studies of yeast exposed to 6-hydroxylaminopurine or AID/APOBEC cytosine deaminase. Selected mutants in mutation rate assays had substantially higher mutation burdens than non-selected isolates from the same population[15]. Episodic APOBEC mutagenesis also occurs in human cell cultures propagated for prolonged periods[16]. Moreover, limited single-cell propagation of human cancer cell lines coupled to whole-genome sequencing (WGS) revealed broader than expected variation in mutation rate in closely related subclones[17]. Observations such as these challenge the assumption that the mutation rate is constant and beg higher resolution studies of mutator cells.

The asymmetrically dividing budding yeast, S. cerevisiae, is ideal for studying mutator phenotypes with high resolution. It encodes many of the same DNA replication and mismatch repair genes found in humans. Yeast "daughter" cells can be separated from their larger "mother" cell at each division by micromanipulation and then moved to defined locations on an agar plate, forming a "single-cell lineage". WGS of cultures derived from these cells permits the number of new mutations that arose in the mother cell at each division to be counted. Moreover, the small size of the genome (12 megabases) makes it cost effective to score enough cell divisions to see whether the distribution of mutation counts conforms to that expected from a single underlying mutation rate.

We previously pioneered this approach with haploid mutator mother cells deficient in Polε proofreading and MMR (pol2-4 msh6Δ)[18]. A single underlying mutation rate could not explain the distribution of mutation counts from 87 divisions. However, the distribution did fit a model with two underlying mutation rates that differed by 10-fold (0.4 and 4 mutations/genome/division). This led to a hypothesis of "mutator volatility" in which cells assumed one of two mutator states as they passed through the cell cycle[18]. But since we only scored mutations retained by the mother, we could not exclude an alternative hypothesis: that polymerase errors sporadically segregated asymmetrically between mother and daughter cells, either as mismatches at the initial division or as permanent double-stranded mutations following the next round of synthesis. Here, to distinguish between these two hypotheses, we sought to score all replication errors that arose in individual cell divisions using more extensive single-cell lineages. Examination of the distribution of the full

replication error counts from individual divisions provided a way to test the mutator volatility hypothesis apart from the confounding influence of segregation. At the same time, sequencing complete lineages gave us the means to determine whether replication errors segregate asymmetrically. The full replication error counts from two different mutator genotypes produced unimodal distributions that were significantly overdispersed relative to that expected from a single underlying mutation rate. Our data suggest that mutator volatility in these cells derives from continuous variation in the underlying mutation rate. Moreover, we found that asymmetric segregation due to the normal process of semiconservative DNA replication and mitotic segregation of chromosomes further expands the distribution of new mutations in individual mutator cells.

## Results

**Evidence for mutator volatility.** To confidently score replication errors arising on all nascent DNA strands from each division, we devised a scheme that ensured that all mutations were observed in at least two members of a single-cell lineage. After moving each daughter by micromanipulation from the founding mother cell, we isolated a sublineage of three additional cells to help score the number of errors segregated to that daughter. These cells included the first and second granddaughter (born to the daughter cell) as well as the first great-granddaughter cell derived from the first granddaughter (Fig. 1a). Errors segregated to the daughter as mismatches in the first division segregate as double-stranded mutations in the next division when the daughter produces the first granddaughter. Mutations retained by the daughter after that segregation event will be inherited by the second granddaughter, forming what we call the "Da" segregant group. Mutations segregated to the first granddaughter will be inherited by the great-granddaughter, forming the "Db" segregant group. In theory, the Da and Db segregant groups represent half of the errors made by the mother cell during a given division. The remaining errors, retained initially by the mother as mismatches, segregate between the mother and her next daughter as double-stranded mutations in the next division. The mutations segregated to that daughter will be uniquely present in the next sublineage, forming the "Ma" segregant group. Mutations retained by the mother will be found in all later sublineages, defining the "Mb" segregant group. After colony formation and WGS, a full error count for a given division can be determined by simply summing the number of mutations in the Da, Db, Ma, and Mb segregant groups. With a complete set of sublineages from the same mother cell, the full replication error counts from several sequential cell divisions can be determined from the nested data (Fig. 1b). By requiring that all errors be observed in at least two members of the lineage, this approach eliminates false positives due to sequencing errors or clonal sweeps within the cultures.

We initially began our experiments with the pol2-4 msh6Δ haploid strain used in the previous study[18]. We found evidence for a more limited mutator volatility but were concerned that lethality within some sublineages may have introduced a bias (see Supplementary Information and Supplementary Fig. 1). To improve the viability and the mutational signal, we switched to using diploid yeast with a 10-fold higher mutation rate due to homozygous mutations affecting Polδ proofreading and base-base mismatch repair (pol3-01/pol3-01 msh6Δ/msh6Δ)[18,19]. To obtain pol3-01/pol3-01 msh6Δ/msh6Δ cells, we mated pol3-01 msh6Δ haploids, freshly dissected from sporulated POL3/pol3-01 MSH6/msh6Δ diploids. We isolated the newly formed zygotes and then used the first or second diploid daughters as founding mother cells for the isolation of single-cell lineages, noting the time and placement of each cell. Following colony formation, and WGS, we

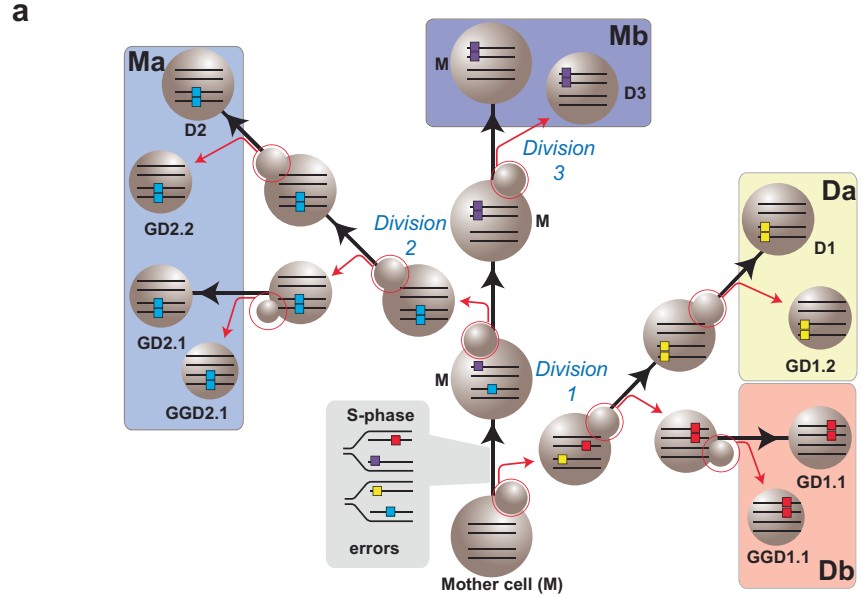

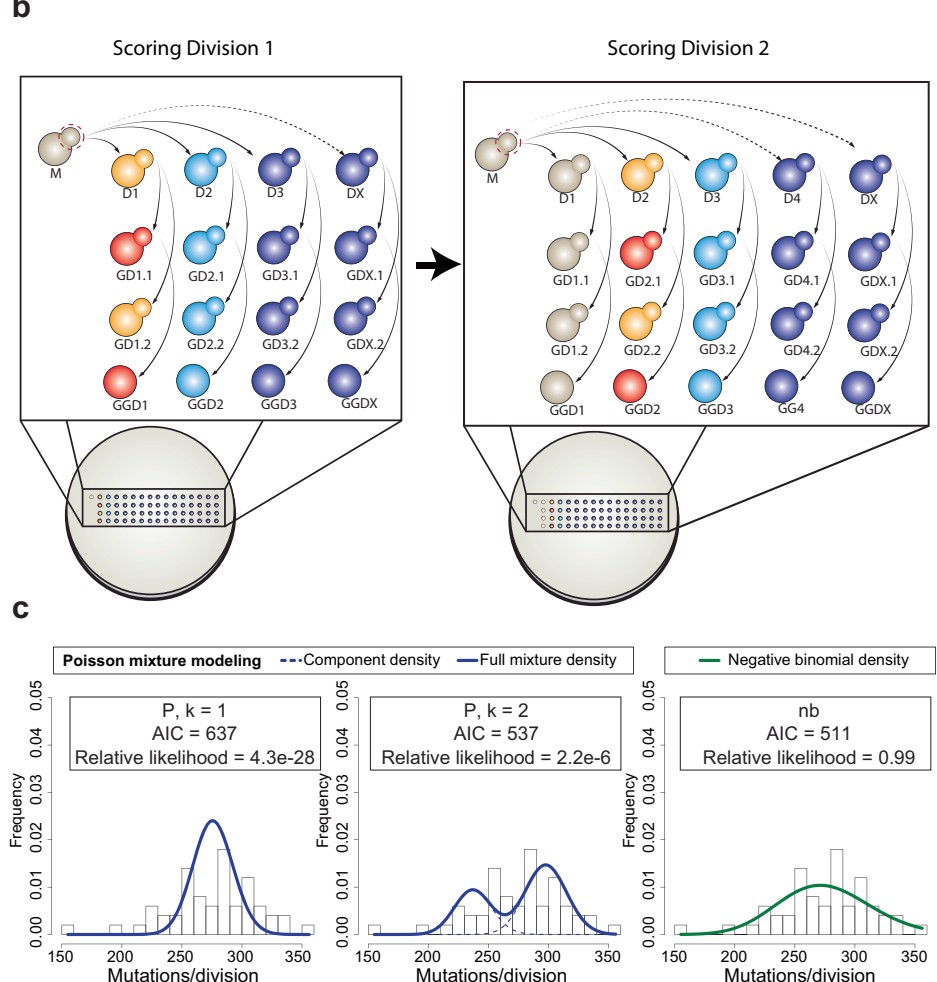

scored 13,801 mutations from 50 divisions obtained from seven different lineages (Fig. 1c, Supplementary Table 1, Supplementary Fig. 2). The mutations were distributed across the genome and displayed a spectrum consistent with combined proofreading and MMR deficiency (Supplementary Fig. 3). We only scored mutations at genomic sites confidently called in all members of a lineage and carefully vetted the resulting variant lists. Having complete lineage information allowed us to assign when the mutations arose using the logic described above. In addition, we visually inspected the variant sites in all genomes from a given lineage using the Integrative Genomics Viewer, which allowed us to detect discrepancies in the lineage order or whether mutations

**Fig. 1 Mutator DNA polymerase errors at single-cell resolution. a** Isolation of single-cell pedigrees. Using microdissection, the founding mother (M), daughter (e.g. D1), granddaughter (e.g. GD1.1, GD1.2) and great-granddaughter (e.g. GGD1.1) cells from each maternal division ($n = 50$) are separated (red arrows) and moved to isolated regions on the plate to form colonies, which are then sequenced. Polymerase errors arising during the initial S-phase are passed on to four segregant groups, highlighted by large colored boxes (Da, Db, Ma, Mb), the sum of which represents the full error count for that division. Large spheres connected by black arrows represent the same cell through multiple divisions. Small spheres circled in red represent budding daughter cells; parallel lines in cells, double-stranded DNA; colored boxes on lines, polymerase errors. **b** Scoring full error counts from sequential divisions. Arrows depict movement of each dissected daughter cell and their descendants to unique positions on the plate to form sublineages. Color-coding indicates cells that will form colonies used for the Da (yellow), Db (red), Ma (blue), and Mb (purple) segregant groups in a given division (See AH121 in Extended Data Fig. 2 for an example). Segregant group identities shift one sublineage to the right with each division. **c** Fitting the distributions of full error counts from diploid *pol3-01/pol3-01 msh6Δ/msh6Δ* divisions to different models. $k = 1$, single Poisson; $k = 2$, two-Poisson; nb, negative binomial; AIC, Akaike information criterion.

had been incorrectly assigned (see Methods). We tallied the full replication error counts from each division and determined whether the distribution could be explained by a single underlying mutation rate.

Mutagenesis has been modeled for more than 70 years[19–21] with the Poisson distribution, which is a discrete probability distribution of the number of expected independent events occurring within a defined interval, assuming a constant rate (λ). A simple test of whether a distribution matches a single Poisson is to calculate the index of dispersion ($\hat{D}$), which is equal to the variance of the distribution divided by the mean ($\sigma^2/\mu$). The variance of Poisson distributions always equals the mean, which results in a $\hat{D}$ of 1. The *pol3-01/pol3-01 msh6Δ/msh6Δ* mother cells committed an average of 276 (±37.7, standard deviation (σ)) replication errors per division. This corresponds to a $\hat{D}$ of 5.15 ($37.7^2/276$), which suggests that the distribution does not conform to a single Poisson (Fig. 1c). Two alternative explanations failed to account for the overdispersion. For instance, we did not observe any relationship between the mother's replicative age and the number of errors made by Polδ (Spearman's rank correlation coefficient: 0.007209, $p = 0.9604$)(Supplementary Fig. 4), nor did the number of mutations correlate with the size of the scored genome, which differed between lineages due to variation in sequencing depth and the number of members in each lineage (Spearman's rank correlation coefficient: −0.0416, $p = 0.7743$)(Supplementary Fig. 4). Instead, the broad distribution of full replication error counts, free from the confounder of segregation, is consistent with mutator volatility.

To better understand the nature of mutator volatility in *pol3-01/pol3-01 msh6Δ/msh6Δ* cells, we used finite mixture modeling, which employs a maximum likelihood framework to identify mixtures of two or more Poisson distributions that better fit the data. We also modeled the data as a negative binomial (nb), which is a discrete distribution with a separate rate (μ) and shape parameters (θ) commonly used to interpret overdispersed count data. The rate parameters λ and μ, for the Poisson and nb distributions, both define the mean number of events. Since these models derive from different distributions, they cannot be directly compared using standard statistical tests. Non-nested models such as these can be evaluated with Akaike Information Criteria (AIC), which uses maximum likelihood to estimate the loss of information of each model relative to the observed distribution. To prevent overfitting, AIC penalizes models with more parameters. Lower AIC values correspond to a more parsimonious fit; however, interpreting the difference in the magnitude of raw AIC values is not intuitive. Thus, we transformed the raw AIC values to "Akaike weighted values", which convey their relative likelihood (Fig. 1b)[22,23]. We found that the negative binomial model was the most likely (relative likelihood of 0.9999), followed by the two-Poisson-mixture model ($2.2 \times 10^{-6}$), and the single Poisson ($4.3 \times 10^{-28}$) (Fig. 1c). Similar results were obtained using Bayesian Information Criteria (BIC), which imposes stronger penalties for overfitting. Thus, mutator volatility

in *pol3-01/pol3-01 msh6Δ/msh6Δ* cells is more complex than just two distinct mutator states.

**Mutation rate varies between divisions**. The superiority of the negative binomial model suggests that the mutator phenotype may vary continuously. This rationale derives from the ability to describe a negative binomial as a gamma-Poisson distribution (Fig. 2a). The gamma function is a continuous, rather than discrete, distribution. Here, it takes the same shape parameter (θ) as the negative binomial and serves as a conjugate-prior to define variation in the rate parameter λ of a mixture of Poisson distributions. The variation in λ that creates a negative binomial occurs between replication events at the same site, or a collection of sites such as a chromosome or genome. Having complete lineage information provided an opportunity to test whether λ varies at a chromosomal or genome-wide level. The distributions of mismatches segregated to mother (Mm) or daughter cells (Dm) across all divisions were the same and fit a negative binomial (Fig. 2b). If λ varied widely during the replication of individual replicons (the units of DNA replication on a chromosome), this could introduce asymmetry in the number of errors on sister chromatids, which would then propagate to the daughter and mother cells (Fig. 2c). Consequently, Dm and Mm from the same division would be free to vary within the observed negative binomial distribution. Alternatively, if the genome-wide value for λ varies between cell divisions, a single mutation rate would govern mismatch formation for both the mother and daughter genomes (Fig. 2d). Dm and Mm would co-vary within the constraints of the corresponding Poisson distribution. To distinguish between these two hypotheses, we first compared the correlation of mismatches segregated to mother and daughter cells to simulated data generated under the constraints of the two models. While no correlation was seen between Dm and Mm in the simulated data from the "replicon variant" model ($R^2 = 0.001$), similar correlations were observed for both the simulated data from the "division variant" model ($R^2 = 0.47$) and the actual data ($R^2 = 0.37$). This correspondence in the number of mismatches segregated to mother and daughter cells extended down to the level of chromosomes (Fig. 2e). The $R^2$ values are lower than typically seen with strong correlations, but as our modeling shows, this is expected since both $X$ and $Y$ values are randomly drawn from a Poisson distribution. As a second test of the hypotheses, we also performed 10,000 simulations of how each model would affect the distribution of full replication error counts from 50 divisions (Fig. 2f). With the replicon variant model, the simulated index of dispersion (3.28 ± 0.66, σ) was substantially less than observed with the actual data ($\hat{D} = 5.15$), while the division variant model produced a good match (5.54 ± 1.12, σ). Together, these analyses strongly suggest that the source of mutator volatility is variation in the genome-wide mutation rate from one division to the next.

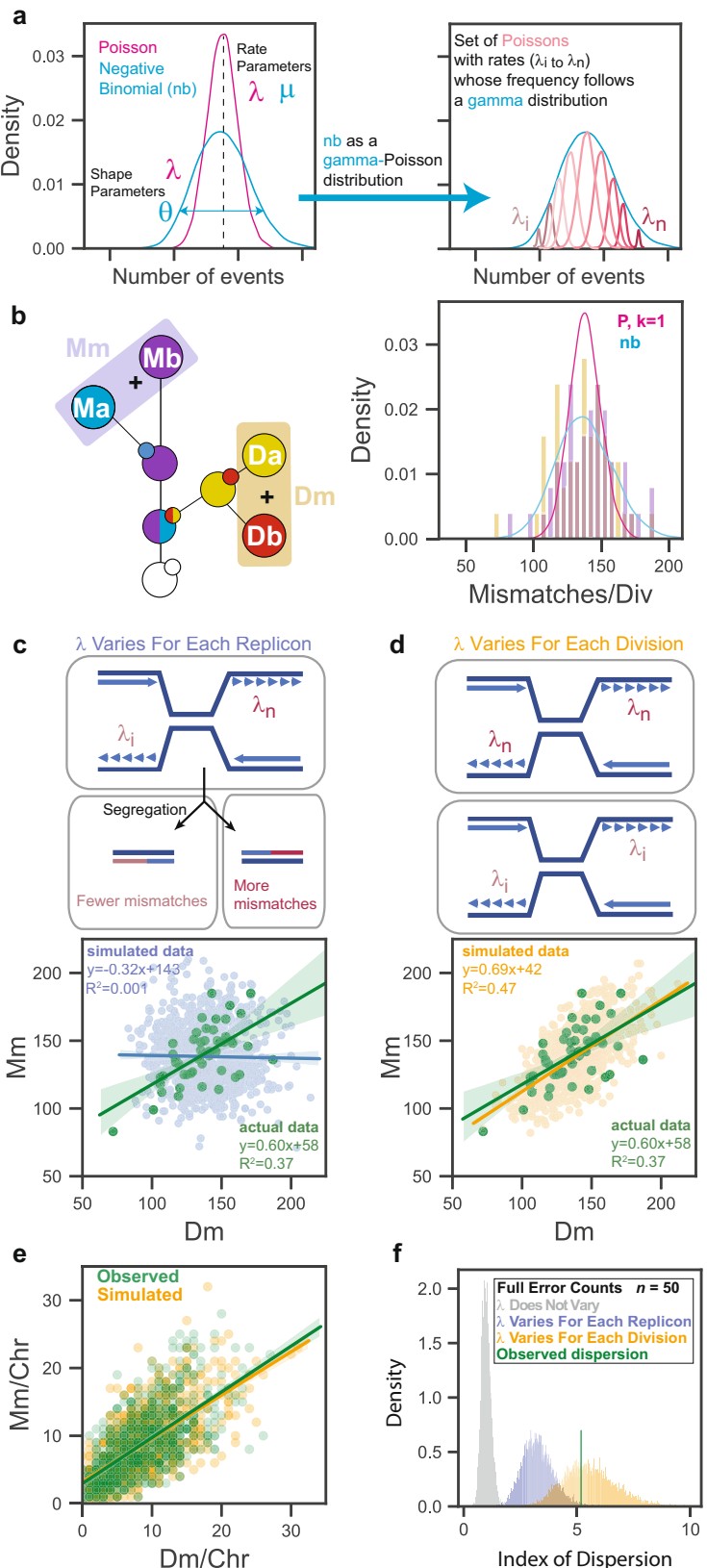

**Asymmetric inheritance of new mutations**. With this support for the mutator volatility hypothesis, we turned our attention to the question of asymmetric inheritance of new mutations. Individual cells averaged 69 (±18, σ) new mutations/diploid genome/division (n = 200) (Fig. 3a) with an index of dispersion of 4.8. A negative binomial fit the distribution most closely (relative likelihood = 0.82), followed by a four-Poisson mixture model (relative likelihood = 0.18). A close examination of mutations arising from the same division revealed a striking asymmetric pattern of inheritance. When pairs of segregant groups were compared

**Fig. 2 Evidence that mutation rate varies between divisions. a** The negative binomial as a gamma-Poisson distribution. The gamma distribution takes the same shape parameter ($\theta$) as the negative binomial and describes the variation in the rate parameter ($\lambda$) of a continuous mixture of Poisson distributions. **b** Schematic of single-cell lineage showing summing of segregant groups to determine the number of mismatches segregated to the mother (Mm) or daughter (Dm) in a single division. Actual distributions are represented by gold (Dm) and purple (Mm) bars. Lines depict models of data: pink, single Poisson (P, $k = 1$); aqua, negative binomial (nb). **c, d** Correlations between Mm and Dm counts from actual data (green, $n = 50$) and simulations ($n = 1000$) under two different models. In (**c**) top panel depicts a cell with converging replication forks from two replicons with different mutation rates. Bottom panel shows the correlation of simulated Mm and Dm values (blue) drawn from the full negative binomial and their linear regression. In (**d**), top panel depicts two cells replicating DNA with different mutation rates. Bottom shows the correlation of simulated Mm and Dm values (orange) and their linear regression. **e** Correlation between the number of mismatches per chromosome segregated to Mother (Mm) or Daughter cells (Dm). green, observed counts; orange, simulated counts from model in (**d**). **f** Simulated index of dispersion of full replication error counts from small cohorts ($n = 50$) assuming the models from (**c** and **d**).

(e.g. Da vs Db or Ma vs Mb), half of the time one segregant group inherited all of the mutations for a given chromosome while the other received none (Fig. 3b, c). This pattern is explained by the sequential actions of semiconservative DNA replication and mitotic segregation of chromosomes (Fig. 3d). At the end of the first S-phase, due to semiconservative replication, all errors arising due to the Poisson process of polymerase error formation reside on one of the two strands of each sister chromatid. These strands segregate equally between mother and daughter cells. The next round of replication produces two new duplexes per cell, only one of which contains double-stranded mutations. At metaphase, cells receive either all or none of the new mutations for that chromosome from the previous division. This binomial process occurs twice for every chromosome number in diploid cells. Consequently, for each chromosome number, cells receive 0%, ~50%, or 100% of the mutations in a given division with a "Mendelian" ratio of 1:2:1 (Fig. 3c) (actual ratio, 876:1490:834). Thus, we can describe how polymerase errors arise in an individual division and later become permanent as a compound Poisson-binomial process.

To determine the contribution of the Poisson-binomial process to the overdispersion of mutation counts, we simulated mutagenesis in *pol3-01/pol3-01 msh6Δ/msh6Δ* cells assuming a constant error rate. Given that we observed an average of 138 mismatches per diploid mother or daughter cell (Fig. 2b), the average rate of error formation was 69 errors/haploid genome/ division. Since cells only inherit, on average, half of the polymerase errors, the observed mutation rate in *pol3-01/pol3-01 msh6Δ/msh6Δ* cells was 34.5 mutations/haploid genome/ division. To model the Poisson-binomial process we simulated mutagenesis on each chromosome by setting $\lambda$ equal to 69 errors/ haploid genome and then, to mimic segregation, multiplied the number of mutations apportioned to each chromosome by a randomly chosen 1 or 0, before summing the total mutations (Fig. 3e). For comparison, we simulated mutation accumulation assuming a simple Poisson process in which mutations accumulated with a rate of 34.5 mutations per haploid genome (Fig. 3e). With 1000 simulations of 200 cell cohorts, the Poisson-binomial model produced a broader index of dispersion ($\hat{D} = 3.58 \pm 0.49$, $\sigma$) than the Poisson model ($\hat{D} = 1.0 \pm 0.1$, $\sigma$) (Fig. 3f), but narrower than the observed data ($\hat{D} = 4.8$). However, substituting the constant mutation rate with the gamma-distributed set of $\lambda$ values from Fig. 2c yielded simulated data with an equivalent dispersion ($\hat{D} = 4.80 \pm 0.49$, $\sigma$) (Fig. 3f). Thus, the combination of mutator volatility and asymmetric segregation of mutations—a gamma-Poisson-binomial process—accounts for the observed distribution of mutations in individual *pol3-01/pol3-01 msh6Δ/msh6Δ* cells.

To understand the potential implications of our findings for mutator-driven cancers, we first focused on how the Poisson-binomial process would influence the heterogeneity of mutation burden within a dividing population of tumor cells. Assuming a

constant mutation rate comparable to *pol3-01/pol3-01 msh6Δ/ msh6Δ* yeast, the expected distribution of simulated mutation counts in human cells after one division ($\hat{D} = 50$) was far broader than in yeast (Fig. 3g) and persisted through 30 simulated divisions (Fig. 3h, i). Adding a comparable level of volatility to the mutator phenotype further increased the simulated dispersion ($\hat{D} = 82$) (Fig. 3g). Using the Poisson-binomial model, we simulated a range of mutator phenotypes observed in cancer cells and found a linear relationship between mutation rate and predicted index of dispersion. For instance, mutation accumulation in HCT116, the well-known MLH1 mutant colon cancer cell line, increases from 48 to 190 mutations/haploid genome/division upon introduction of a heterozygous *POLE* proofreading-deficient allele[9]. In these cells, the predicted index of dispersion expanded from 3.4 to 10.8 (Fig. 3j). Even greater heterogeneity may arise in human cancers when more potent *POLE* mutator alleles occur in combination with MMR deficiency[5,7,24,25]. Thus, the fundamental Poisson-binomial process of asymmetric segregation of new mutations has the potential to dramatically expand the diversity of mutation burdens present among a population of human mutator cells.

## Discussion

Genetic heterogeneity progressively increases in a dividing population of cells as an unavoidable consequence of errors made during DNA synthesis. Here, for the first time, we describe the fate of polymerase errors made on all nascent DNA strands synthesized in individual cell divisions. We developed this single-cell resolution approach in order to understand previous observations that the distribution of new mutations in individual mutator cells was broader than expected. To explain the phenomenon, we proposed two hypotheses: (1) that mutator phenotypes are volatile and (2) that polymerase errors arise with a constant rate but segregate asymmetrically on the way to becoming double-stranded mutations. The design of our single-cell pedigrees ensured at least two independent biological observations for each mutation, which allowed us to confidently assign more than 13,000 mutations to fifty divisions. From the resulting mutation count data, we found strong evidence that both mutator volatility and asymmetric segregation of new mutations significantly expand genetic heterogeneity in *pol3-01/pol3-01 msh6Δ/ msh6Δ* yeast.

Historically, mutagenesis has been modeled with the Poisson distribution, which describes the probability of the number of independent events per unit time given a constant rate. The observed distribution of full replication error counts of mutator cells, free from the influence of segregation, best fit a negative binomial and not a single Poisson (Fig. 1c). Negative binomials are equivalent to a continuous mixture of Poisson distributions whose rates vary according to a gamma distribution (Fig. 2a). This suggests that mutator volatility may create a continuum of mutation rates rather than discrete mutator states. We explored

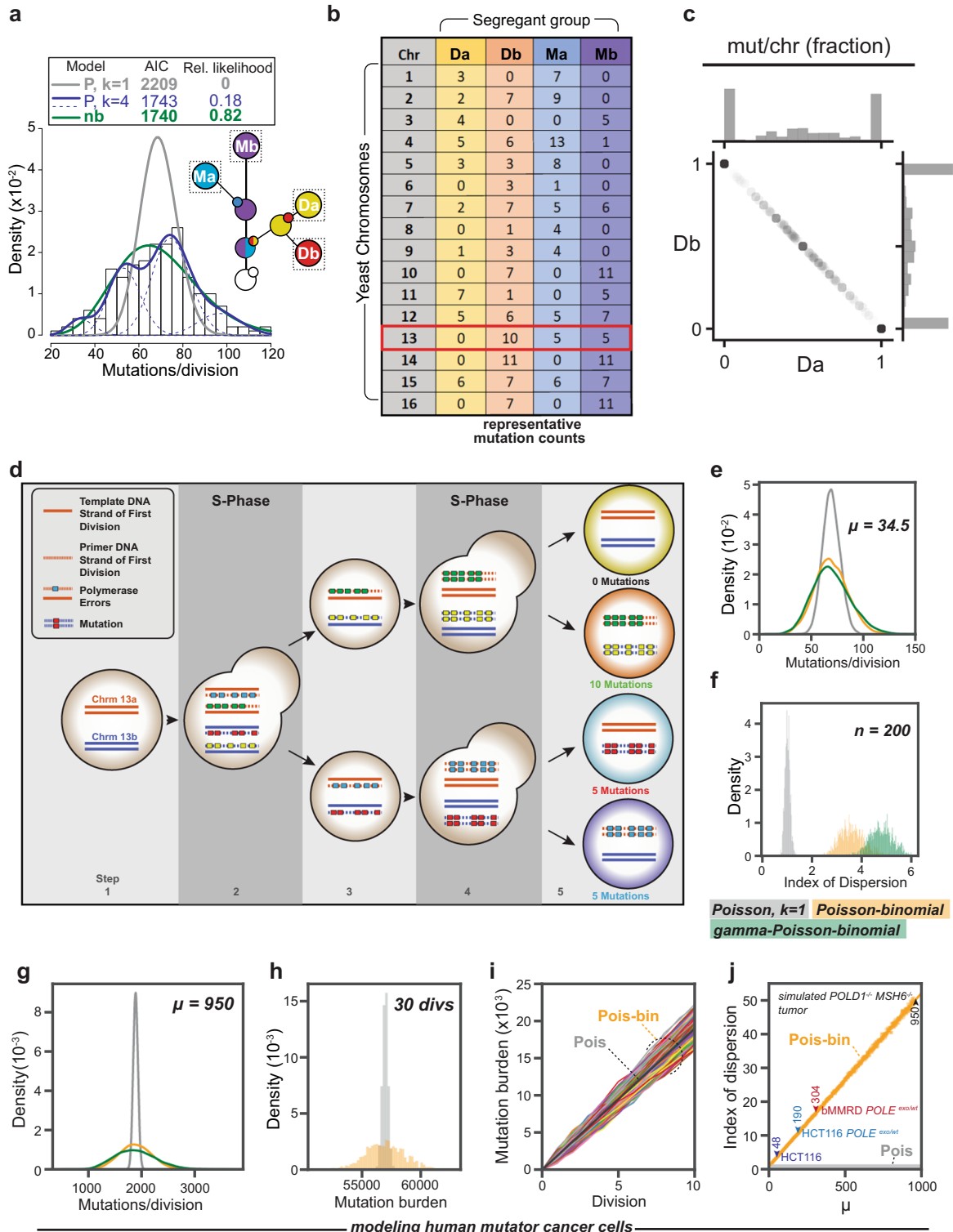

the idea that mutation rate varies from one division to the next by simulating the number of mismatches segregated to mother and daughter cells (Fig. 2d, e) and the dispersion of full replication error counts expected from small cohorts of cells (Fig. 2f). Both simulations closely matched the observed data, consistent with the hypothesis that mutator volatility derives from continuous variation in mutation rate between divisions. Two caveats are worth noting. First, this model of volatility was developed from a relatively small sample size ($n = 50$). Substantially increasing the number of scored divisions may reveal that the underlying

distribution is derived from discrete mutator states rather than a continuum of rates. Second, in cells with different mutator alleles the underlying distribution may vary substantially depending on how the mutator alleles interact with the currently undefined source of volatility, and mutator phenotypes affecting other processes besides DNA replication may have different sources of volatility. Mutator polymerases do not operate as a closed system. They interface with a myriad of other replication components and metabolites, such as dNTPs, that influence their fidelity[26,27]. Variation in the timing and duration of perturbations to these

**Fig. 3 Asymmetric segregation broadens the distribution of mutation burden in mutator cell populations. a** Combined distribution of new mutations arising in the Da, Db, Ma, and Mb segregant groups (see inset) from *pol3-01/pol3-01 msh6Δ/msh6Δ* lineages ($n = 200$). Key of models (top): gray line, single Poisson (P, $k = 1$); blue lines, four-Poisson (P, $k = 4$), green line, negative binomial (nb). AIC, Akaike information criterion. **b**, Table of representative mutation counts from one division of a diploid mutator cell. Columns represent different segregant groups; rows, the chromosome (chr) number; values, the total number of new mutations found on homologous chromosome pairs. Red box indicates a chromosome with both asymmetric and equal sharing of mutations. **c** Segregation of double-stranded mutations between Da and Db. For each division, the fraction of mutations observed in Da or Db on each chromosome was determined and then plotted against each other. **d** Asymmetric segregation of new mutations: (1) Two homologous chromosomes in mother cell (orange and blue lines) prior to scored division. (2) During the first S-phase, mutator Pol δ generates errors (colored boxes) in the nascent strands (dashed lines). (3) At segregation, mother and daughter each inherit two chromosomes with mismatches. (4) In the next S-phase, strands with mismatches produce new double-stranded mutations, while error-free strands do not. (5) Segregation results in cells with 0, 1, or 2 chromosomes with new double-stranded mutations. **e**, Simulated distributions of mutations/division at a rate of μ = 34.5 ($n = 10,000$) assuming a single Poisson process (gray), a Poisson-binomial process (orange), or a gamma-Poisson-binomial process (green). **f** Variation in the index of dispersion of simulated data from the 3 models ($n = 200$) over 1000 iterations. **g** Simulated distribution of mutations/division in human ultra-mutator cells assuming a mutation rate ($μ = 950$, $n = 10000$) comparable to *pol3-01/pol3-01 msh6Δ/msh6Δ* yeast and a single Poisson process (gray), Poisson-binomial process (orange), or gamma-Poisson-binomial process (green). **h** The cumulative mutation burden of a human ultra-mutator cell after 30 simulated divisions with (orange) and without (gray) asymmetric segregation. **i** Simulated trajectory of mutation burden of human mutator tumor cells (Colored lines, $n = 1000$) undergoing a Poisson-binomial process compared to a Poisson process (black line). **j** Change in the index of dispersion under a Poisson-Binomial process (orange line) compared to the static index of dispersion under a Poisson process (gray line at bottom) with an increasing mutation rate. Colored markers represent estimated mutation rates for clinically relevant mutator-driven HCT116-derived mammalian cancer cell lines[9] and a tumor from a patient with biallelic MMR deficiency (bMMRD)[34].

interactions may produce volatility. The observed overall mutation rate that cells exhibit represents a composite of mutation rates at all sites within the genome. Conceivably, the change in replication fidelity could be localized to certain parts of the genome in a given division. But if so, our data suggest, that the nascent strands from each pair of sister chromatids in the affected region must be equally influenced by the change in rate (Fig. 2c, f).

The asymmetric inheritance of new mutations observed in mutator cells results from the fundamental processes of semi-conservative replication and mitotic segregation of chromosomes acting in concert. Current models of mutation accumulation generally ignore the potential for this synergy to expand genetic heterogeneity, although there are exceptions. John Cairns proposed a far more extreme asymmetric inheritance of mutations in the "Immortal Strand Hypothesis" in which stem cells always segregated away newer DNA duplexes with double-stranded mutations[28]. In keeping with this hypothesis, a recent computational analysis of human somatic variants argued that the high variance of mutation burden in adult stem cells with age supports a preferential inheritance of ancestral strands[29]. A second study from the field of evolutionary biology examined the potential influence of disparate mutagenesis of leading and lagging strand synthesis to promote variable evolutionary trajectories from the same cell population[30]. Our findings here demonstrate that, in the context of a mutator phenotype, the *normal* process of semi-conservative replication and mitotic segregation of chromosomes has the potential to create unequal sharing of mutations. We find no evidence that daughter or mother cells preferentially inherit new mutations (Supplementary Fig. 5). For every cell that inherits disproportionately more mutations there will be another cell with fewer mutations. The predicted impact of this process on the variation in mutation burden is larger in human cells than in yeast due to the vast differences in chromosome length, and the correspondingly larger number of errors per chromosome. However, with longer chromosomes comes an increased likelihood that sister chromatid exchanges (SCEs) may mitigate the asymmetry. SCEs clearly to do not homogenize mutation burden in diploid mutator yeast cells since half of cells either received all or none of the new mutations for a given chromosome (Fig. 3c). A high frequency of SCEs would have left few chromosomes with 0 mutations. This finding is in keeping with recent evidence from a sensitive Next Generation Sequencing methodology (Strand-seq) that SCE occurs with a rate of 0.26 events/division in yeast[31]. Strand-seq experiments of normal human fibroblasts and lymphoblasts indicate the SCEs occur with a rate of 5 events/cell division[32]. At this rate, most chromatid pairs in mutator cells would be free of SCEs even after the two divisions it takes for polymerase errors to become double-stranded. Of course, the frequency of SCEs may increase in some cancer cells, especially those with certain intrinsic DNA repair defects[32]. Performing single-cell lineage analysis of human mutator cells in future studies should address both the prevalence of SCEs and the asymmetric inheritance of mutations.

Our simulation of a mutator-driven tumor rapidly generated substantial intra-tumoral genetic heterogeneity during expansion (colored lines, Fig. 3i) compared to a population in which mutations accumulated by a simple Poisson process (black line, Fig. 3j). The associated variability in mutation load may be relevant to cancer evolution. Early during tumorigenesis the subpopulation of cells that inherit disproportionately more mutations may adapt more readily. With elevated mutation rates, polyclonal adaptation is almost certain. The unifying feature of these adapted cells is a high mutation burden. As mutation burden mounts and mutator cells contend with increasingly strong negative selection pressure due to immune surveillance and negative epistatic interactions[33,34], adapted cells that inherit fewer new mutations due to asymmetric inheritance may be at a relative fitness advantage. In this context, selectively increasing mutation rate in mutator cancer cells could represent a novel therapy[26]. If, as a means of treatment, the mutation rate of cancer cells is only transiently elevated to induce extinction, this subpopulation may persist. Sustained elevation of mutation rate over many divisions of mutator cells may be required to drive their extinction.

## Methods

**Yeast strains and culture conditions**. The diploid strains AH2801 (*POL2/URA3:: pol2-4 MSH6/msh6Δ::LEU2*)[18] and AH2601 (*POL3/URA3::pol3-01 MSH6/msh6Δ:: LEU2*)[35] are derived from AH0401, a BY4743 derivative engineered to be heterozygous at the *CAN1* locus (*CAN1::natMX/can1Δ::HIS3*) to facilitate forward mutation rate assays[33]. We followed standard procedures for yeast propagation and tetrad dissection[36]. For general propagation, we grew liquid YPD cultures (1% wt/vol yeast extract, 2% wt/vol peptone, 2% wt/vol dextrose) at 30 °C. For sporulation, we diluted overnight YPD cultures 1:100 in 3 mls of YPD and grew until the culture reached $1–2 \times 10^7$ cells/ml. We recovered the cells by centrifugation, resuspended and pelleted the cells once in 1 ml H$_2$O, and then resumed growth at 22–25 °C in 2 mls of sporulation media (1% potassium acetate, 0.1% yeast extract,

0.05% dextrose) for five days. For rich solid media, we used synthetic complete (SC) [6.7 g Difco yeast nitrogen base without amino acids, 2% wt/vol dextrose, 2 g/L SC amino acid Mix (SCM) (Bufferad)] supplemented with 2% wt/vol agar. For plates lacking leucine and uracil (SC-Leu-Ura), SCM was substituted for SCM-Leu-Ura (Bufferad). Archival frozen stocks were stored in 23% glycerol at −80 °C.

**Single-cell lineage isolation**. To isolate pol2-4 msh6Δ lineages we dissected AH2801 tetrads on SC-Leu-Ura selective media and chose one germinating spore per plate to serve as the founding mother cell. To obtain pol3-01::URA3/pol3-01::URA3 msh6Δ::LEU2/msh6Δ::LEU2 cells for pedigree analysis we first dissected POL3/ pol3-01::URA3 MSH6/ msh6Δ::LEU2 tetrads on SC-Leu-Ura plates. After two divisions, double mutant haploid cells from different tetrads were placed next to each other to allow mating. Upon isolation of a zygote, the first or second daughter was used as the founding mother (M) for the lineage. Mothers were placed at an isolated location and we separated daughter cells (designated Dn, Dn+1, etc.) from the mother as they were generated and moved them to select areas 5 mm apart on the plate. We repeated the procedure to obtain each daughter's first daughter (GD.1, Fig. 1b), second daughter (GD.2), and first granddaughter (GGD, born to GD.1). This strategy was repeated for each daughter up to either the 20th division or the end of the mother's replicative lifespan, whichever occurred first. In a typical experiment, we pre-punched the agar with the dissecting needle at each drop-off location so that we would always put the cell in a defined place, making it easy to later find the cell for inspection and manipulations. We isolated lineages over the span of a week by performing rounds of dissections every 90–120 m. Only a few cells on a plate were moved in any one round, and then, only one cell at a time. We noted the timing of each round of bud dissections. We incubated plates at 30 °C between dissections. At the end of the day, plates were wrapped in parafilm and stored overnight at 4 °C. When plate dissections were concluded, we incubated each plate an additional 48 h at 30 °C to allow colonies to fully develop. Prior to sequencing, the pol3-01/pol3-01 msh6Δ/msh6Δ and pol2-4 msh6Δ genotypes were confirmed by allele-specific PCR assays[35].

**Genome sequencing**. Each colony in a pedigree was used to inoculate overnight 5 ml liquid YPD cultures for WGS[35]. Glycerol stocks were made and genomic DNA extracted with the ZR Fungal/bacterial purification kit (Zymo Research). DNA was sheered into 500 to 1000 bp fragments by sonication. After end-repair, Illumina sequencing libraries were made by ligating on dsDNA adapters and indexing by quantitative PCR. The samples were then sequenced on the HiSeq 2500 or Nextseq platforms. We performed sequencing alignments and variant calling using a custom pipeline (eex_yeast_pileline.sh) that runs in the Unix command-line (see Github link in Code Availability). Reads were aligned to a repeat-masked S288C yeast genome[18] using the Burrows-Wheeler Aligner (0.7.17)[37]. We removed discordant and split-read groups using Samblaster (0.1.24)[38]. We used Picard tools (2.21.9) AddorReplaceReadGroups to add information to the header used for later steps in the analysis. We then indexed the BAM files with Samtools (1.8)[39]. To minimize false variant calls, we sequentially processed the BAM files with functions from the Genome Analysis Toolkit (GATK3)[40] including RealignerTargetCreator, IndelRealigner, LeftAlignIndels, BaseRecalibrator, and PrintReads. We made a pileup file with Samtools and used VarScan (v2.3.9) mpileup2snp to call single nucleotide variants[41]. We limited our analysis to single nucleotide variants, which are by far the most abundant polymerase error type in these cells. We used the Varscan2 tool to identify variants present in our colonies with the following parameters. For pol2-4 msh6Δ haploid lineages we used a variant frequency cut-off of 0.8 with a minimum read depth of 18 (daughter and GD.1 positions) or 10 (for GD.2 and GGD positions). Since these are haploid cells, new variants should be present in 100% of reads. Setting the cut-off at 0.8 accommodates sites with low read depth and one sequencing error. For pol3-01/pol3-01 msh6Δ/msh6Δ diploids, we used a minimum read depth of 18 for all strains and a variant frequency cut-off of 0.22. With a read depth of 18, clonal heterozygous variants in diploid cells have a false negative rate of 6.1 ×10⁻⁵. With 1000 mutations we have a 6% chance of having 1 false negative in a genome. We filtered the above results to remove variants present in the parental strains as well as recurrent sequencing artifacts. A small number of variants (<0.1%) could be reliably scored with the above parameters but fell below a quality threshold for a subset of genomes. These were manually curated for inclusion. We detected these by visually inspecting the BAM files for all strains in a single-cell lineage at the same time using the Integrated Genome Viewer (IGV).

**Scoring of mutations and detection of assignment errors**. We used a custom Python script (JLSLineageCaller) to determine the number of shared variants within each lineage. The program first determines all genomic positions with 18-fold read depth in all members of the lineage and then filters the called variant lists for mutations at positions within the shared genome. Pairwise comparisons are done between certain strains to identify shared mutations at different branch points in the lineage, resulting in a data-frame of comparisons that allows all mutations arising in a lineage to be sorted and examined in Microsoft Excel. The mutation counts for division n were determined by summing the number of new mutations identified at branch points Da (GDn.1 vs GGDn.1), Db (Dn vs GDn.2), Ma (Dn+1 vs GDn+1.1), and Mb (Dn+2 vs Dn+3). Da mutations are only found in the

daughter (Dn) and her second daughter (GDn.2). Likewise, Db mutations are only found in GDn.1 and her first daughter GGDn.1. Mismatches retained by the mother after the first division become double-stranded mutations in the next division and are either passed on to her next daughter (Dn+1) or are retained by the mother and passed on to all future offspring. The mutations inherited by Dn+1 that form the Ma segregant group are only found in this branch of the lineage. Finally, the mutations retained by the mother, the Mb segregant group, first appear in Dn+2 and her offspring, but also show up in all subsequent daughters (Dn+3, Dn+4, etc) and their offspring. Any deviation from this pattern of inheritance indicates an "assignment error" has occurred and that a cell was inadvertently placed in the wrong position in the lineage. In the Supplementary Information, we describe two such cases. The divisions encompassing these strains were censored from the analysis. Below we describe how these errors arise and are detected to illustrate the reliability of the method.

One possible assignment error could occur at dissection when the daughter and mother cells both divide before the next round of dissection. On the basis of size, the first daughter (Dn) can be easily distinguished from the mother, the second daughter (Dn+1), and her own daughter (GDn.1). Usually Dn+1 and GDn.1 can also be distinguished because Dn+1 buds before GDn.1. However, in rare cases Dn+1 and GDn.1 are adjacent and similarly sized. If Dn+1 is moved in place of GDn.1, we will have a sublineage consisting of Dn, Dn+1, GDn.2, and GDn+1.1 (instead of Dn, GDn.1, GDn.2, and GGDn.1). Every sublineage would normally contain subsets of mutations from different divisions (Da and Db mutations from the "n" division; Ma mutations from the "n-1" division; and Mb mutations from the "n-2" division). In this sublineage, the Ma segregant group mutation count will be 0, since there are no new mutations that will be shared by these four colonies. However, a substantial subset of the mutations assigned to the Db segregant group will also be found in later sublineages indicating that they are not Db mutations but Mb mutations from a later division. The other half of what appear to be Db mutations will in fact be Ma mutations from a different division. Added confirmation of the dissection error comes from the analysis of the next sublineage, which will consist of GDn.1 (not Dn+1 as it should be), GGDn.1, GGDn.2, GGGDn.1 (great-great-great granddaughter 1). There will be 0 Mb mutations in this sublineage since all of these cells are directly descended from Dn. These problematic cell divisions would be censored because we lack key lineage members necessary to obtain a full replication error count. Another type of assignment errors could occur during dissections to isolate the sublineages. For instance, if Dn divides twice in the interval before the next round of dissection we would have to distinguish between GDn.1 and GDn.2. This is usually easy to do because, as above, GDn.1 would be forming a bud while GDn.2 would be unbudded. If we inadvertently reversed those two cells, we would have a sublineage consisting of Dn, GDn.2, GDn.1, and a great granddaughter born to the second granddaughter. When calling the Da segregant group we would be calling shared mutations between Dn and GDn.1 (and not between Gn and GDn.2). We would quickly see that these are, in fact, Ma segregant group mutations because they would also be present as a subset of Db mutations obtained in the comparison between GDn.2 and her offspring.

The most difficult potential assignment errors to detect would occur in the Da and Db segregant groups. For example, if GDn.1 divided twice, producing GGDn.1 and GGDn.2, and we selected GGDn.2 instead of GGDn.1, the mutation count for the Db segregant group would be derived from two divisions instead of one. Again, this is unlikely, because GGDn.1 would begin budding long before GGDn.2. But we lack an obvious distortion to the pattern of mutation inheritance to flag this as an error. We don't think this is a common problem given the correspondence between mismatches segregated to the mother (Mm) and daughter (Dm) cells illustrated in Fig. 2d,e. As described above, we regard the Ma and Mb segregant groups as highly reliable because dissection errors lead to obvious perturbations in the pattern of mutation inheritance. In favor of the reliability of the Da/Db data, an XY scatter plot of mutation counts observed in pairs of Ma/Mb segregant groups corresponds very well to that observed with pairs of Da/Db segregant groups (Supplementary Fig. 5). Both sets also correspond with what would be expected based on simulated data. (The simulation assumed a gamma-Poisson distribution as in Fig. 2). Interestingly, there are two Ma/Mb (47,36) and Da/Db (52, 20) segregant pairs in the lower left-hand quadrant that appear as outliers. Both pairs are derived from Division 15 (Supplementary Data 1), leading to the conclusion that the mutation rate in that division was inherently low. The highest Da/Db outlier (51,120), derived from Division 8, is also associated with a Ma/Mb pair with high mutation counts (120,65), leading to the conclusion that this division had a high mutation rate.

**Statistical modeling**. We grouped the mutation counts from the above branch points into Da, Db, Ma, and Mb segregant groups to determine their distributions. We also joined all segregant groups into one larger group to examine the distribution of mutation counts across all cell divisions. To determine the distributions of mismatches segregated to the daughter (Dm) and mother (Mm) cells, we first summed the Da and Db or Ma and Mb mutation counts from each division. We also combined these two sets into one group to view the distribution of mismatches across all cell divisions. To determine the distribution of total polymerase errors per division, we summed all mutations from individual divisions (Da+Db+Ma+Mb). We considered two common approaches for modeling overdispersed

count data: the Poisson mixture distribution and the negative binomial distribution.

A $K$-component Poisson mixture distribution, which we denote PM($K$), has a probability mass function (pmf) given by

$$f_{\text{PM}}(x; K, \mathbf{p}_K, \boldsymbol{\lambda}_K) := \sum_{k=1}^{K} p_k f_{\text{Poisson}}(x; \lambda_k), \qquad (1)$$

where $\mathbf{p}_K = (p_1,\ldots,p_K)$ is a vector of mixture proportions, $\boldsymbol{\lambda}_K = (\lambda_1,\ldots,\lambda_K)$ is a vector of Poisson means, and $f$ is the pmf of a Poisson($\lambda_k$):

$$f_{\text{Poisson}}(x; \lambda_k) := \frac{\lambda_k^x\, e^{-\lambda_k}}{x!}. \qquad (2)$$

From this formulation, we see that the full density of the distribution is decomposed as a sum of the scaled Poisson densities. In (1), $p_k$ represents the prior probability that a given count measurement will be generated from the $k$th Poisson component distribution, parameterized by $\lambda_k$. Since a given count measurement could have been generated from any of these $K$ components, we average over their densities based on their prior probabilities to get the full density of that count.

The negative binomial distribution can be specified by the following probability mass function:

$$f_{\text{NegBinom}}(x; \mu, \theta) := \frac{\Gamma(x + \theta)}{x!\Gamma(\theta)}\left(\frac{\theta}{\theta + \mu}\right)^\theta \left(\frac{\theta}{\theta + \mu}\right)^x, \qquad (3)$$

where $\mu$ is the rate parameter and $\theta$ is the shape or dispersion parameter. As $\theta$ tends towards zero, the variance increases. As $\theta \to \infty$, the negative binomial reduces to a Poisson distribution.

We implemented these principles using a single R script (FMM.R, see Github link below). To fit Poisson mixture models we used the flexmix R package in R v3.5.3[42]. To fit negative binomial models we used the glm.nb function of the MASS R package[43]. Goodness of fit testing of the models was performed using both Akaike information criterion (AIC) and Bayesian information Criterion (BIC) in R. Although these two approaches score fit in slightly different ways, BIC returned results consistent with AIC and we thus report only the more commonly used AIC scores. We scored each tested distribution against up to four parameters. We reported only up to the number of parameters that improved model fit. Lower raw AIC values indicate better fit; however, the relative differences are not immediately intuitive and so we calculated Akaike weighted values[21,22]. To illustrate this approach, the AIC values in Fig. 1b were 637, 537, and 511. The first step in getting weighted AIC values is to determine $\Delta_i$AIC: the difference between each AIC value and the AIC with the lowest value (so for these numbers: 126, 26, 0). The likelihood of each is then calculated by $\exp(-1/2 \times \Delta_i\text{AIC})$. The weighted AIC value for a given model is its likelihood divided by the sum of all competing likelihoods. From these calculations the weighted AIC values are 4.3e−28 (P, $k$=1), 2.2e−6 (P, $k$=2), and 0.9999978 (nb), respectively. Thus, the negative binomial model is far more likely than the other two models to account for the observed data. Mixture model graphs were constructed using the ggplot2 package R[44]. Spearman rank correlation coefficients were calculated using the Scipy Stats package in Python and graphs generated with Seaborn 0.9.

**Simulation of negative binomial models**. We wrote a Python script (Fig2.py, see Github link below) to simulate the expected correlation between Dm and Mm under two distinct models of mutagenesis (Fig. 2). The script uses the $\theta$ (60.42) and $\mu$ (138) parameters estimated by glm.nb for the negative binomial model of mismatches segregated to mother (Mm) or daughter (Dm) cells (see FMM.R). (Note that glm.nb actually returns the natural log value for $\mu$ (in this case 4.927), which must be exponentiated ($e^{4.927}$) to get 138). In the first model, we assumed that the negative binomial distribution was created by variation in mutation rate along chromatid pairs, so that upon segregation, Dm and Mm from the same division were free to vary within the predicted negative binomial distribution. To simulate this process with Scipy.stats.nbinom.rvs, we converted the $\theta$ and $\mu$ shape parameters to the n and p inputs (see script for details) for nbinom.rvs and then, for each division, we selected two random values from the distribution to represent the Dm and Mm counts. In the second model, we assumed that the negative binomial was created by a gamma distribution of $\lambda$ values for a series of Poisson processes acting in different cell divisions. We used Scipy.stats.gamma.rvs to simulate $\lambda$ values from a gamma distribution with shape and scale parameters derived from those of the negative binomial. The shape parameter for the gamma distribution is simply equal to $\theta$. With variance (v) equal to $\mu^2/\theta$, the scale parameter is equal to $v/\mu$. With a random $\lambda$ from the gamma distribution as an input for Scipy.stats.poisson.rvs, we selected two values from the associated Poisson distribution to serve as Dm and Mm counts for each division. To examine the relationship between Dm and Mm in these different models and the actual data, we performed linear regression with Scipy.stats.linregress and visualized the data and regression line using Seaborn 0.9 regplot.

**Simulation of gamma-Poisson-binomial process**. We wrote Python scripts to create a Poisson-binomial model of the contributions of semiconservative DNA replication and mitotic segregation to the overdispersion of mutations in individual yeast (Fig3ef.py, ExFig6.py) and human cells (Fig3ghij.py) depicted in Fig. 3 and

Supplementary Fig. 6. For yeast simulations, we determined the amount of unmasked DNA on each chromosome in the repeat-masked genome and then divided these values by the total length of unmasked DNA in the haploid genome. The rate of mismatches per haploid genome (69 mismatches/haploid genome/ division for *pol3-01/pol3-01 msh6Δ/msh6Δ* cells) was then multiplied in each case by these fractions to obtain the per chromosome rate of mismatch formation. These values were used as input for scipy.stats.poisson.rvs to simulate the number of errors per chromosome in a single division. We created two independent entries per chromosome to model the diploid genome. To mimic the binomial process of mitotic segregation, we then multiplied the number of simulated errors on each chromosome by a randomly chosen 1 or 0. Finally, we summed the mutation counts from all chromosomes to obtain the total number of new mutations per cell division. To create a gamma-Poisson-binomial model, we selected a value for lambda at each division from the gamma distribution described in Fig. 2 rather than using a constant rate for mismatch formation. As a control we performed the same simulation without the binomial process, using the mutation rate per haploid genome (34.5 mutations/haploid genome/division). We used the same approach for the human simulations except that we multiplied the fraction of each human chromosome of the total genome (GRCh38) by a mismatch rate comparable to that observed with *pol3-01/pol3-01 msh6Δ/msh6Δ* yeast: 69 mismatches/ haploid yeast genome/division × ($3.03 \times 10^9$ bp/human haploid genome/$11 \times 10^7$ bp/yeast haploid genome) = 1900 mismatches/human haploid genome/division. We compared the resulting distribution to that from a Poisson distribution with $\lambda$ equal to 950 mutations/haploid genome. To simulate the diversity in mutation burdens that this process generates, we summed the simulated mutation counts for individual lines from 30 divisions.

## Data availability

Sequence data used to generate the findings of this study have been deposited in the NCBI Sequence Read Archive (SRA), BioProject accession: PRJNA586886. Source data for graphs and charts can be found in Supplementary Data 1 and Supplementary Data 2. All data files are available upon request to the corresponding author.

## Code availability

Scripts used to generate figures and perform statistical tests have been deposited to github: https://github.com/idowsett/Asymmetric-segregation-of-polymerase-errors-and-rate-volatility-diversify-mutation-burden (DOI: 10.5281/zenodo.4272887).[45] A custom sequence analysis bash script eex_yeast_pileline utilizes established tools including BWA (0.7.17), Samblaster (v.0.1.24), Picard-tools (2.2.2), Samtools (1.8), GATK (4.0.6.0), Varscan (2.3.9), and a Python 2 script (Variant_deSNPer2013a.py) written to remove single nucleotide polymorphisms in our strain background from the variant calls. A Python 2 script (JLSlineage_caller.py) calls shared variants in our lineages. Figures and Supplementary figures generated using code are provided as standalone Python 3.6.5 or R v3.5.3.

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

## Acknowledgements

The authors would like to thank Larry Loeb, Sage Malingen, and Marina Watowich for their critical review of our manuscript. This study was supported by the National Institute for General Medical Sciences (NIH/NIGMS R01GM118854). ITD was supported by the UW Molecular Medicine Training Grant (NIH/NIGMS T32GM095421), the Genetic Approaches to Aging Training Grant (NIH/NIA T32AG000057), and the Biological Mechanisms for Healthy Aging Training Grant (NIH/NIA T32AG066574). The content is solely the responsibility of the authors and does not necessarily represent the official views of the NIH, NIA, or NIGMS.

## Author contributions

I.T.D., S.R.K., and A.J.H. designed research; I.T.D., J.S., J.M., E.M., and A.J.H. performed research; I.T.D., J.S., B.J.O., and A.J.H. analyzed data; I.T.D., and A.J.H. wrote the paper.

## Competing interests

The authors declare no competing interests.
