## [Peer Review File · Communications Biology]

Reviewers' comments:

Reviewer #1 (Remarks to the Author):

The manuscript describes accurate measurements of genome-wide mutation rates in individual cell divisions of diploid yeast carrying a combination of mutator alleles inactivating proofreading in lagging strand polymerase delta (pol3-01) and partially inactivating post-replicative mismatch repair (msh6). Authors evolved their experimental methodology (initially published in PLOS Genetics, 2015) to eliminate artifacts and increase measurement accuracy. As a result, they accumulated data that, beyond the questions raised and resolved by this manuscript, can serve as a base line for mutation rates and spectra resulting from DNA polymerase errors. Importantly, the elaborate experimental approach of sequencing fresh single-cell derived colonies reduces the impact of selection to the theoretical minimum. It would be interesting to compare mutation spectrum accumulated due to polymerase errors collected in this work with the spectra obtained in multiple bottleneck passages of mutator yeast in a work from the other lab (PMID 25217194). Authors may choose to add this analysis to the manuscript to broaden its impact.

Accurate measurement of mutation numbers acquired in individual cell divisions allowed authors to support their hypothesis of cell to cell variation of the numbers of mutation events is due to yet unknown physiological fluctuations mutation rates as well as to asymmetrical segregation of DNA strands in a cell progeny. The range of variation and the shape of distribution can be used for predicting cell to cell variation in vegetative populations of mutator cell, including hyper-mutated and ultra-mutated tumors.

There are several points of presentation improvement that I recommend.

1. The term "mutator cells" is not commonly used and may be misleading, especially for people reading a title, because it implies active role of some cells in causing mutations in other cells. What about "cells with mutators alleles" or shorter "cells with mutators"?
2. line 12 - Replace "tandem defects" with "combined defects"
3. lines 17-18. Replace "broader than expected for a single rate " with "broader than expected for mutation rate uniform across all cells"
4. I am not sure if "Mendelian segregation" used across the manuscript is applicable here. It usually implies germline. What about "chromosome mitotic segregation".
5. line 33. "ultra-hypermuted" genomes". Ultramutated and hypermutated are applied to different kinds of tumors with high mutation load; see e.g., how these classifications are presented in TCGA-based PMID: 31811476
6. Lines 39-40. Reference to Petljak et al PMID: 30849372 on episodic mutagenesis in cancer cell lines would be useful here
7. Starting from line 173. :first "model" - give names to each of the two models and use these names rather "first" and "second" in discussing rather complex statistical evidence
8. "However, the mutator yeast strains studied here do not show obvious signs of

elevated SCEs” – explain the reasons for such a statement. There was no assessment of SCEs in this work.

9. Figure 1b – X-axes in each of the three graphs.

"mutation score" is not defined in any place of the manuscript. In reviewing, I interpreted it as a count of mutation events in a single replication. Please, define.

10. Supplementary datasets 1 and 2. Add column “chromosome” to the tabs “Full Mutation List”. Currently chromosomes could be only deduced by going to GenBank, which is inconvenient.

Each “Lineage” tab contains two tables apparently repeating each other. Detailed legends for these tabs would make them useful for those that would like to use the data for addressing another hypotheses or as a baseline.

Reviewer #2 (Remarks to the Author):

In this study, the authors recorded all mutations arise from a serial of cell divisions. The data they collected allow them to test several hypotheses. Including the correlation between replication age and mutation rate, the immortal strand hypothesis, and their mutator volatility hypothesis. In this paper they focused on the mutator volatility hypothesis, they found that a model considering variable mutation rate and asymmetric segregation best fit the observed distribution. Overall, this is a valid study. I have a few questions listed below.

1. Is there a proposed molecular mechanism behind the “mutator volatility” phenomenon? The genes knockout in this current study is different from their previous study, so the processes from which mutations arise are also different, how would this affect the mutator volatility?

2. The experiment is based on a diploid cell, but the illustration in Fig1A appears to be haploid. Also, de novo mutations are not immediately fixed in diploid cells, they should have a frequency of 50% as they arise. I would recommend the authors to change the illustration to diploid in Fig1A and rephase accordingly in the first paragraph of the Results.

3. The authors mentioned the “Immortal Strand Hypothesis” in the Discussion. The data they generated offers a perfect test of this hypothesis. I wish the authors could do a test, like, in each division, how many mutations is kept by the mother cell, and how many went to the daughter cell.

Reviewer #3 (Remarks to the Author):

The process by which tumors evolve and accumulate mutations has been discussed for decades. This manuscript addresses a long-standing question; is there a linear increase in the number of mutations or are there mechanisms whereby different cell divisions renders varying amounts of mutations. Specifically, the authors attempt to determine to what extent polymerase errors, that later are fixed as mutations, are asymmetrically segregated between mother and daughter cells, and whether the

number of polymerase errors vary from cell cycle to cell cycle due to unknown reasons. To accomplish this, the authors studied a diploid yeast strain completely lacking Pol delta proofreading activity and with a reduced capacity to repair mismatches (pol3-01/pol3-01 msh6Δ/msh6Δ cells). In my opinion a wise choice, since a much larger number of mutations can be scored compared to a haploid strain or a strain with pol epsilon proofreading deficiency. After that, single cell lineages after each cell-division was whole genome sequenced. The identified mutations are analyzed and compared to mathematical models. The authors conclude that there is a volatility in genome-wide mutation rates between each cell-division. Furthermore, they also find evidence for asymmetric segregation of mutations. Combined these two mechanisms provide an opportunity for sudden increases of mutations in daughter cells in tumors. Such sudden increases of mutations in cells may drive the development of cancer and give growth advantages to new cell lineages within a tumor. Something that was postulated by Larry Loeb a very long time ago. In addition, I agree with the authors that at later stages may the asymmetric segregation of replication errors be beneficial for the survival of the tumor in the sense that daughter cells that inherit less mutations are better fit for survival due to the overall mutation burden.

Overall, this is a very exciting paper!

We offer below a point by point response (in blue) to the comments of the reviewers and explain how we used those to improve the manuscript.

Reviewer #1 (Remarks to the Author)

There are several points of presentation improvement that I recommend.

1. The term "mutator cells" is not commonly used and may be misleading, especially for people reading a title, because it implies active role of some cells in causing mutations in other cells. What about "cells with mutators alleles" or shorter "cells with mutators"?

It is true that "mutators" originally were used to refer to alleles or mobile DNA elements. But there are important examples in the literature of using the phrase "mutator cells" to denote cells with a mutator phenotype. This can be found in the writings of Jeffery Miller (UCLA) (see PMID: 11371538 and PMID: 10049391) as well as Andrew Murray (Harvard) who offers the following definition:

"We refer to cells and populations that contain such mutations as mutators and to their counterparts that lack these mutations and have lower mutation rates as nonmutators. (PMID: 16920619)".

Both investigators have made important contributions to our understanding of mutator evolution. The journal *Science* even used the term in a title of a paragraph highlighting an article in one of their issues: "Capturing a Mutator Cell"
<https://science.sciencemag.org/content/288/5465/397.8>.

Nevertheless, we appreciate, based on the reviewer's comment, that this could be confusing for some people and so have revised it. Moreover, in the Introduction we define the term "mutator cell" so that there will be no need for confusion for our readers when we use the term later in the manuscript.

2. line 12 - Replace "tandem defects" with "combined defects".

Revised.

3. lines 17-18. Replace "broader than expected for a single rate " with "broader than expected for mutation rate uniform across all cells"

Revised

4. I am not sure if "Mendelian segregation" used across the manuscript is applicable here. It usually implies germline. What about "chromosome mitotic segregation".

The point is well-taken. Of course, the principles that Mendel observed in Meiotic segregation informed our understanding of mitotic segregation. We have revised the manuscript accordingly using "mitotic segregation of chromosomes" or just "mitotic

segregation”.

5. line 33. “ultra-hypermuted” genomes”. Ultramutated and hypermutated are applied to different kinds of tumors with high mutation load; see e.g., how these classifications are presented in TCGA-based PMID: 31811476

The term “ultra-hypermuted” goes back to the first papers describing these tumors. “Ultramutated” has certainly superseded it and we have revised our manuscript accordingly.

6. Lines 39-40. Reference to Petljak et al PMID: 30849372 on episodic mutagenesis in cancer cell lines would be useful here.

We have added this reference.

7. Starting from line 173. :first “model” - give names to each of the two models and use these names rather "first" and "second" in discussing rather complex statistical evidence.

We have revised the text using the terms “replicon variant model” and “division variant model”.

8. “However, the mutator yeast strains studied here do not show obvious signs of elevated SCEs” – explain the reasons for such a statement. There was no assessment of SCEs in this work.

Earlier in that same paragraph we say:

"SCEs clearly do not homogenize mutation burden in diploid mutator yeast cells since half of cells either received all or none of the new fixed mutations for a given chromosome (Fig. 3c)."

To emphasize the point we have added,

“A high frequency of SCEs would have left few chromosomes with 0 mutations.”

To further explain our thinking, in Fig. 3c, we show the fraction of mutations per chromosome inherited by either of the members of a segregant pair (Da or Db). We calculated this for each Da-Db pair by dividing the number of fixed mutations per chromosome in Da or Db by the total number of mutations per chromosome (Da + Db). Opposite the X and Y axes are histograms that show the abundance of the different fractions. There are three general classes: (1) Db or Da inherit all of the mutations (fraction = 1), Db and Da share the mutations ($0 < \text{fraction} < 1$), or Da or Db inherited none of the mutations (fraction = 0). The frequency of these three classes exhibits a perfect 1:2:1 “Mendelian” ratio. If significant SCE had occurred, the intermediate class would increase at the expense of the “all” or “none” classes. This holds true for the Ma and Mb segregant pairs as well.

9. Figure 1b – X-axes in each of the three graphs.

"mutation score" is not defined in any place of the manuscript. In reviewing, I interpreted it as a count of mutation events in a single replication. Please, define.

The reviewer was correct in their interpretation and we have revised "mutation score" to mutations/division in the panels in Figure 1b.

10. Supplementary datasets 1 and 2. Add column "chromosome" to the tabs "Full Mutation List". Currently chromosomes could be only deduced by going to GenBank, which is inconvenient.

We thank the reviewer for the suggestion and have included the chromosome number as a separate column.

Each "Lineage" tab contains two tables apparently repeating each other. Detailed legends for these tabs would make them useful for those that would like to use the data for addressing another hypotheses or as a baseline.

We had used the split function in Excel to facilitate navigating the worksheet. We have removed the split to eliminate this confusion.

Reviewer #2 (Remarks to the Author):

1. Is there a proposed molecular mechanism behind the "mutator volatility" phenomenon?

We speak in general terms about the possible molecular mechanism in the Discussion where we say:

"Mutator polymerases do not operate as a closed system. They interface with a myriad of other replication components and metabolites, such as dNTPs, that influence their fidelity^{25,26}. Variation in the timing and duration of perturbations to these interactions may produce a continuum of rates. The observed overall mutation rate that cells exhibit represents a composite of mutation rates at all sites within the genome. Conceivably, the change in replication fidelity could be localized to certain parts of the genome in a given division. But if so, our data suggests, that the nascent strands from each pair of sister chromatids in the affected region must be equally influenced by the change in rate (Fig.2c,f)."

We are currently testing the hypothesis that perturbations in dNTP pools explain mutator volatility, but don't yet have sufficient data for a conclusion.

The genes knockout in this current study is different from their previous study, so the processes from which mutations arise are also different, how would this affect the mutator volatility?

We do, in fact, describe experiments with the *pol2-4 msh6Δ* genotype in the Supplementary Section that we used in our previous study (which we reference in the main body of the Results). We found evidence for a modest volatility consistent with what we observed in the much stronger mutator strain. The index of dispersion was 2.2 and the distribution best fit a negative binomial (relative likelihood = 0.75). We agree that volatility could be more or less pronounced with other mutator alleles.

2. The experiment is based on a diploid cell, but the illustration in Fig1A appears to be haploid.

The reviewer is correct. We wanted to show how mutations segregated on two different chromosomes and depicted it as haploid for simplicity. We realize now that this may be confusing when in the context of the data in Fig2B.

Also, de novo mutations are not immediately fixed in diploid cells, they should have a frequency of 50% as they arise.

We borrowed terminology used by Cairns (in speaking of the Immortal Strand Hypothesis) to talk about how a mismatch becomes a permanent mutation. We realize that this may be confusing to evolutionary biologists where “fixation” in a diploid context means that a mutation becomes homozygous. We thank the reviewer for catching this.

I would recommend the authors to change the illustration to diploid in Fig1A

We have revised the figure to show two homologous chromosomes in a diploid context, akin to the depiction of asymmetric segregation in Extended Data Fig.6.

and rephase accordingly in the first paragraph of the Results.

Throughout the manuscript we have removed the terms “fix”, fixed, or “fixation” and replaced them with alternatives. At times we use the term “double-stranded mutation”, which was employed in a recent *Nature News and Views* by Graham and McClelland (<https://www.nature.com/articles/d41586-020-01815-6>).

3. The authors mentioned the “Immortal Strand Hypothesis” in the Discussion. The data they generated offers a perfected test of this hypothesis. I wish the authors could do a test, like, in each division, how many mutations is kept by the mother cell, and how many went to the daughter cell.

We agree that our data offers a powerful test of the Immortal strand hypothesis, but only for yeast. Further experiments with asymmetrically dividing human stem cells will be required to fully test the hypothesis. The reviewer wants to know about how mutations are distributed between mother and daughter cells. We do in fact show how new mutations are shared between mother and daughter cells in Extended Data Fig.7. In the previous version we plotted D_a and M_a on the X-axis and D_b and M_b on the Y-axis,

as well as simulated data (segregant 1 vs segregant 2). Thanks to the reviewer's question, we realized it would have been more correct to have daughters on the X-axis (Db, Ma) and mothers on the Y-axis (Da, Mb). The key take-away from our analysis is that there doesn't seem to be a systematic bias in segregation. The regression lines for Db vs Da and Ma vs Mb overlap with that of the simulated data with no bias in segregation. To make this explicit and to draw the reader's attention to Extended Data Fig.7, we now include the following sentence:

"We find no evidence that daughter or mother cells preferentially inherit new mutations (Extended Data Fig. 7)."

REVIEWERS' COMMENTS:

Reviewer #1 (Remarks to the Author):

Revised version properly addressed my concerns and critique

Reviewer #2 (Remarks to the Author):

I appreciate the authors' efforts spending on this revision. All of my previous concerns are properly addressed. I recommend acceptance for publication. Now the illustrations in Figure 1 and Extended Data Fig. 6 seem clear and correct to me.

Reviewer #4 (Remarks to the Author):

I want to congratulate the authors. This was a very interesting and enjoyable read. The hypothesis of over-dispersed mutation accumulation rates and mutational volatility between single cell divisions and across chromosomes is very interesting. I think the manuscript is quite advanced and well explained and I would suggest publication at this stage.

Given the advanced stage of the manuscript and the previous round of revisions, I will not suggest or ask the authors for major revisions or adjustments, I think the paper is fit for publication as it is.

Maybe the authors could add a small comment on the data quantity and possibly in follow up studies a higher resolution would potentially allow to resolve even more details of the mutation rate volatilities.

A second comment could be added on potentially very interesting experiments with non-POLE mutants. I understand that mutation rates are lower in such cells, but it would be interesting to see, if mutation volatility (possibly at a lower level) remains in such more normal cells.

Below we provide a point by point response to the comments of our reviewers in blue.

Reviewer #1 (Remarks to the Author):

Revised version properly addressed my concerns and critique

We are gratified to hear that. Thanks for your valuable insight.

Reviewer #2 (Remarks to the Author):

I appreciate the authors' efforts spending on this revision. All of my previous concerns are properly addressed. I recommend acceptance for publication. Now the illustrations in Figure 1 and Extended Data Fig. 6 seem clear and correct to me.

We are gratified to hear that improvements satisfy your concerns.

Reviewer #4 (Remarks to the Author):

I want to congratulate the authors. This was a very interesting and enjoyable read. The hypothesis of over-dispersed mutation accumulation rates and mutational volatility between single cell divisions and across chromosomes is very interesting. I think the manuscript is quite advanced and well explained and I would suggest publication at this stage.

Given the advanced stage of the manuscript and the previous round of revisions, I will not suggest or ask the authors for major revisions or adjustments, I think the paper is fit for publication as it is.

Maybe the authors could add a small comment on the data quantity and possibly in follow up studies a higher resolution would potentially allow to resolve even more details of the mutation rate volatilities.

We appreciate the suggestion and have included several sentences in the second paragraph of the Discussion to this effect (new lines in blue).

“...Both simulations closely matched the observed data, consistent with the hypothesis that mutator volatility derives from continuous variation in mutation rate between divisions. Two caveats are worth noting. First, this model of volatility was developed from a relatively small sample size ($n = 50$). Substantially increasing the number of scored divisions may reveal that the underlying distribution is derived from discrete mutator states rather than a continuum of rates. Second, in cells with different mutator alleles the underlying distribution may vary substantially depending on how the mutator alleles interact with the currently undefined source of volatility, and mutator phenotypes affecting other processes besides DNA replication may have different sources of volatility. Mutator polymerases do not operate as a closed system...”

A second comment could be added on potentially very interesting experiments with non-POLE mutants. I understand that mutation rates are lower in such cells, but it would be interesting to see, if mutation volatility (possibly at a lower level) remains in such more normal cells.

We agree that these experiments may be interesting. Such cells may exhibit volatility for entirely different reasons – a point we now make in the second paragraph of the discussion. The challenge with using mutator cells with much lower rates of mutation is that the information in the distributions will be heavily zero-inflated. Still, it would be quite remarkable if hypermutable cell divisions were observed alongside many divisions with no mutations. The search for such divisions could be justified if whole genome sequencing of Canavanine-resistant mutants from such yeast populations exhibit much higher mutation burdens than non-selected single-cell isolates from the same population (see Yada et al (2013)).